# From Guardrails to Compilers: A Constraint-First Runtime Substrate for Clinical AI Agents

Kaiping Zheng
National University of Singapore
Singapore
dcszkai@nus.edu.sg

Horng-Ruey Chua
National University Hospital
Singapore
mdcchr@nus.edu.sg

Si Wei Kheok
Singapore General Hospital
Singapore
kheok.si.wei@singhealth.com.sg

Kee Yuan Ngiam
National University Hospital
Singapore
kee_yuan_ngiam@nuhs.edu.sg

Marcus Chun Jin Tan
National University Hospital
Singapore
ophcjmt@nus.edu.sg

Robert John Walsh
National University Cancer Institute
Singapore
robert_walsh@nuhs.edu.sg

## ABSTRACT

LLM-powered clinical agents now read electronic health records, retrieve evidence, and draft recommendations inside molecular tumor boards, ICUs, and rare-disease clinics. Recent evaluations are sobering: agents hallucinate dosages and omit standard-of-care steps, induce automation bias even among AI-trained physicians, fall to medical-advice prompt injection at above 90% success rates, and often cannot produce a defensible audit trail when a decision is later contested. We argue this is a *runtime* failure, not a model failure: the substrate on which medical agents execute lacks first-class data primitives for constraint enforcement, provenance tracking, and policy-aware federation. We therefore sketch a constraint-first runtime that compiles every agent action, whether a tool call, a retrieval, a semantic operator, or a final recommendation, into a constrained, provenance-tracked query plan. Semantic integrity constraints, computable subsets of oncology and sepsis guidelines, ontological axioms over standard clinical vocabularies, and federation-aware access policies are expressed in one specification, from which the runtime compiles a plan whose safety, security, and auditability hold by construction. Under this view, the data-management primitives long refined by the community (semantic constraints and operators, semiring-annotated provenance, secure federated query processing, access control, and incremental view maintenance) become the trusted computing base for clinical agents, and contextual intelligence becomes a matter of query planning rather than model scale. We illustrate the design with a molecular-tumor-board example, sketch a four-layer architecture, and lay out the open questions it raises for the data-management community.

**VLDB Workshop Reference Format:**
Kaiping Zheng, Horng-Ruey Chua, Si Wei Kheok, Kee Yuan Ngiam, Marcus Chun Jin Tan, and Robert John Walsh. From Guardrails to Compilers: A Constraint-First Runtime Substrate for Clinical AI Agents. VLDB 2026 Workshop: Biomedical Data Management Systems (BioDMS).

## 1 INTRODUCTION

A molecular tumor board (MTB) at a major cancer center may review many complex cases in a single session under tight time constraints [76]; preparing each patient's dossier requires painstakingly integrating multiple clinical systems, external genomic reports, pathology slides, prior imaging, and treatment histories, together with a rapidly evolving research literature that may not surface the single most relevant trial in time [38, 72]. Elsewhere in the same hospital, an ICU intensivist operates under an even tighter decision cycle: the New York SEP-1 mandate operationalized a three-hour sepsis bundle whose timely completion has been associated with lower mortality [23, 66] , while any decision support that misses a contraindication or triggers a stale alert risks both patient harm and regulatory scrutiny [3, 24]. Both settings sit on the same edge: too much data and too many rules to weigh by hand, too little time to reason from scratch, and the same later question that an auditor or attorney may pose: *which evidence supported this decision, and was the standard of care satisfied?*

LLM-powered clinical agents promise to compress these workflows [8, 17, 48, 85], yet recent evidence is sobering. A 2025 multi-model study reported 91.8% of surveyed clinicians encountering medical hallucinations and 84.7% believing these could harm patients [36]. Omission rates reach 97% on common diagnostic prompts [79], and clinically relevant omissions outweigh hallucinations head-to-head (3.45% vs. 1.47%) [2]. A randomized trial showed that physicians with formal AI-literacy training still suffered a 14-point drop in diagnostic accuracy under plausible but incorrect LLM suggestions [27, 57]. Adversarial evaluations show degraded performance across leading models [52], and medical LLMs are vulnerable to misinformation injected into training or retrieval [32]; one recent study elicited harmful medical advice through prompt injection with a 94.4% success rate, including 91.7% in severe scenarios involving FDA Category-X drugs [42]. Vision-language oncology models exhibit similar vulnerabilities [16], and clinical-LLM-agent benchmark suites have emerged [34, 48]. The dominant AI-community response is *guardrails*: post-hoc filters, retrieval-augmented citations, and self-checks that treat the agent as a black-box text generator. Figure 1 (left) shows the result: a leaky post-hoc filter in front of a black-box LLM, an overflowing alert-fatigue inbox, a stale guideline binder, and a masked prompt-injection adversary slipping through it.

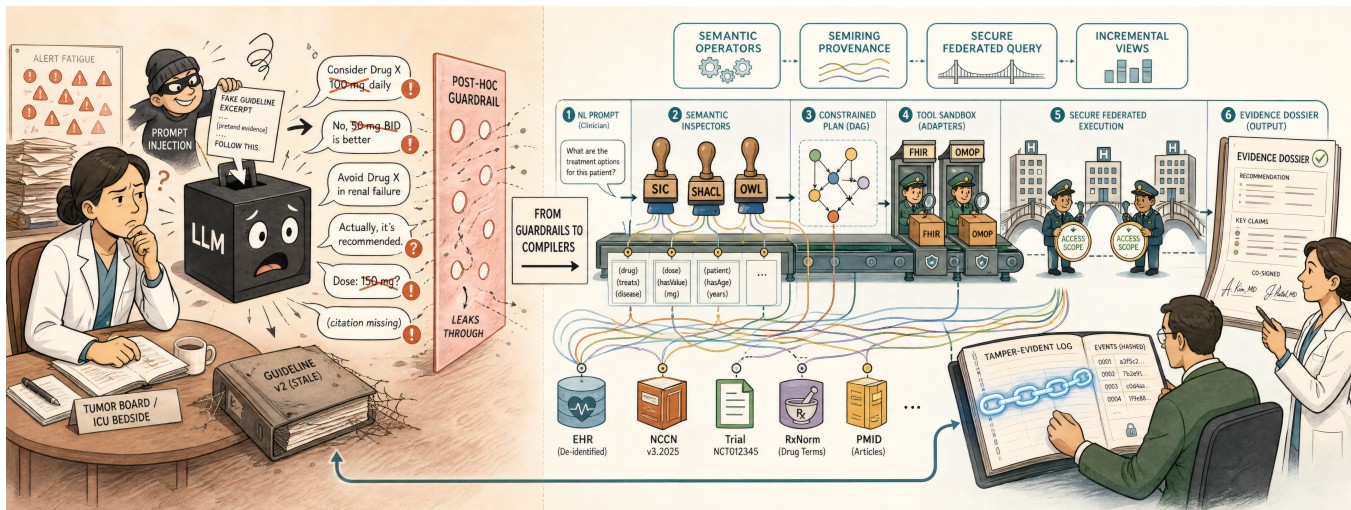

**Figure 1: From guardrails to compilers: the runtime shift this paper argues for.** *Left (today):* a black-box LLM sits behind a leaky post-hoc guardrail; alert-fatigue, a stale guideline binder, and a prompt-injection adversary all slip through. *Right (this paper):* the clinician's natural-language prompt is compiled into a constrained, provenance-tracked query plan: semantic inspectors (SIC, SHACL, OWL) check every step, a sandbox validates each tool output, secure federated execution honors per-institution access scopes over standard sources (EHR, NCCN, trials, RxNorm, PubMed), and a tamper-evident log yields an evidence dossier the clinician co-signs.

The data management community views the same problem differently. *What an agent does is execute a query plan over heterogeneous, governed, and uncertain data.* Under this view, safety and security enforcement belong in the planner and runtime, not after the text. The natural-language prompt is a *compiler frontend* for a constraint-aware physical plan [45, 77]; the LLM is merely one operator among many [25, 43, 47, 51, 53, 75]; and the relevant primitives already exist: semantic integrity constraints [40], fine-grained provenance [13, 14, 29, 56, 62], secure federated query processing [4, 55, 81], access control for data-backed systems [1, 49], and incremental view maintenance over evolving rules [9, 37]. What remains missing is their integration as the trusted computing base of a clinical agent. This reframes *agent security* as a query-optimization and constraint-enforcement problem rather than a text-filtering one. Concretely, it maps clinical-agent safety onto three problems the data-management community has studied for decades: *query optimization* (choosing and ordering operators, now including LLM calls, under a cost model), *integrity-constraint enforcement* (rejecting states that violate declared invariants), and *data integration* across governed, heterogeneous sources (schema mapping, ontology alignment, and access-controlled federation). An agent's sensitivity to the patient context, the active guideline version, and institutional access policy then becomes a compiled, auditable runtime artifact. We call this property *contextual intelligence*: an attribute of the runtime substrate, not of the language model that runs within it.

**Running example.** A tumor-board agent is asked, *"what systemic options does this patient have?"* for a case of metastatic colorectal cancer whose genomic report carries an activating KRAS mutation. A guardrail-wrapped LLM may propose the anti-EGFR antibody cetuximab [35], which is contraindicated for exactly this variant. In our runtime, the same request compiles into a plan: an access-scoped

federated join fetches the patient's variant call and medication list; a semantic operator drafts candidate regimens; a semantic integrity constraint over (`variant`, `drug class`, `recommendation`) fires on the forbidden ($KRAS^+$, anti-EGFR) tuple and rewrites the plan to *warn-and-log* rather than recommend; and every step is stamped into a provenance-tracked dossier. We thread this example through the paper: the constraint appears as class C1 (§2), the primitives that enforce it in §3, and its compilation in §4.

*A threat model for clinical agents.* Four attack and failure surfaces, already observed in deployed prototypes, are structurally database problems. (i) *Prompt injection through retrieved evidence*: a poisoned trial abstract or a manipulated guideline excerpt steers the agent toward an unsafe recommendation [16, 30, 42, 94]. From the planner's perspective, this is untrusted input that must be schema-validated and confined before downstream operators consume it. (ii) *Tool-output corruption*: an upstream electronic health record (EHR) view, a faulty variant call, or a stale knowledge-graph entry introduces an incorrect tuple into the plan. Semantic constraints [40] and constraint-driven cleaning [59] detect the violation before it propagates. (iii) *Unauthorized composition*: an agent's sequence of otherwise permissible tool calls produces a join prohibited by institutional access policies [1, 49], even though no individual call violates policy in isolation. Capability-scoped planning [20] rules out such plans at compile time. (iv) *After-the-fact tampering*: an audit log is modified to retroactively justify a prior decision. Semiring-annotated, append-only provenance [19, 29, 50] renders such edits detectable. Each is what database research has long called an *integrity* problem; medical agents are its most consequential current setting. Figure 1 (right) places each defense on the runtime: stamped inspectors mitigate (i) and (ii), policy-customs at federation bridges defuse (iii), and tamper-evident chained logs address (iv).

**Contributions.** This is a vision paper whose goal is to convince the data-management community that clinical-agent safety is *its* problem to solve. Concretely, we (1) reframe the observed failures of clinical LLM agents as a *runtime* rather than a model deficiency, with a threat model whose four surfaces are classical integrity problems; (2) catalog six classes of constraint that clinical practice already imposes and map each onto a data-management primitive the community has built and validated (§2–§3); (3) sketch a four-layer constraint-first runtime that compiles a unified specification into a constrained, provenance-tracked query plan (§4); and (4) distill the resulting agenda into five open research questions (§5). We do not claim an implemented system; our aim is to fix the abstractions and encourage collaboration.

## 2 CONSTRAINTS FROM CLINICAL PRACTICE

Clinical decisions are made against a large but structured set of rules that today's agents rarely enforce. We organize these rules into six classes of constraint; each is documented in the clinical literature and has a corresponding harm or attack mode. What unifies them is that each is routinely observed in clinical practice yet is enforced today only by human vigilance, not by any runtime invariant on agent action.

**C1: Guideline-derived contraindications and indications.** NCCN, ASCO, ESMO, and Surviving Sepsis guidelines [23, 54] encode subsets of clinical decisions as near-deterministic rules. Our running example is canonical: cetuximab is contraindicated in metastatic colorectal cancer with activating KRAS mutations [35], a forbidden tuple over (variant, drug class, recommendation). Analogous deterministic structure governs HER2-targeted therapy [68], immune-checkpoint blockade in MSI-H/dMMR tumors [39], and KRAS G12C-targeted therapy [67]. ICU sepsis management has its own rule set: the SEP-1 three-hour bundle (lactate, blood culture before antibiotics, broad-spectrum antibiotics, fluid resuscitation), with measurable mortality increase per hour of delay [23, 66]. The computer-interpretable-guideline community has produced multiple formalisms (GLIF [6], PROforma [71], Asbru, GELLO, FHIR Clinical Reasoning + CQL [54, 70]); none integrates with semantic-operator or LLM-call planning at runtime.

**C2: Ontological coding integrity.** Clinical claims must reduce to coding systems the stack can interpret: SNOMED CT for conditions, LOINC for measurements, RxNorm for drugs, and UMLS as the binding glue [5, 11, 83]. Leading LLMs are poor medical coders, mis-mapping concepts at rates unacceptable for structured clinical pipelines [69]. Two failure modes follow. First, cross-domain coding errors (e.g., a measurement coded as a condition) and contradictions across sources [86] pass silently today but should become runtime errors; detecting such data deviations and biases in EHRs is itself an active data-management problem [92, 93]. Second, coding integrity is also a security boundary: free-text labels that are never grounded in an ontology are precisely the opening that prompt injection exploits to reach downstream operators.

**C3: Provenance, corroboration, and tamper-evident audit.** A multi-disciplinary tumor board chair will not co-sign a recommendation whose claims cannot be traced to specific records, guideline sections, and timestamps. Semiring-based provenance [13, 14, 29, 56, 62] provides the natural language, while VeriFact [15] and

DECIDE-AI [80] show concrete clinical demand. The same machinery makes audit-trail tampering detectable via cryptographic hash chaining [19] and provenance-driven APT detection [50].

**C4: Patient-specific longitudinal constraints.** A patient on warfarin must not be prescribed a major-interaction drug; one with eGFR < 30 mL/min/1.73 m$^2$ cannot receive a renally-cleared agent at standard dosing; and one who has declined transfusion must not receive a transfusion-dependent plan. These constraints arise from joins between the agent output and the patient's longitudinal record, not generic facts. A 2024 stepped-wedge cluster RCT showed that ICU CDS alerts *tailored* to patient-specific longitudinal context, rather than generic drug-drug interaction (DDI) rules, reduced administration of high-risk drug combinations, whereas untailored alerts suffered approximately 90% override rates [3, 24, 84].

**C5: Governance and federation policy.** Cross-institutional cohort matching to identify "patients like this" [10, 91] is clinically valuable but constrained by HIPAA Safe Harbor, GDPR, jurisdiction-specific re-identification limits, and institutional data use agreements, forming a non-trivial policy lattice. Multi-institutional networks on the OMOP common data model [11, 58, 83] and secure federated query processing [4, 55, 81] demonstrate legal and technical feasibility, but their integration with semantic-operator pipelines remains ad hoc. An agent crossing institutions in a single trajectory must navigate the corresponding stack of institutional policies [1, 49]. Recent regulation now treats this lattice as legally binding for clinical AI systems, including the EU AI Act for high-risk medical AI [18], FDA Predetermined Change Control Plans [78], and other emerging guidance for non-deterministic clinical software [26, 73], so that policy compliance becomes an auditable obligation rather than a matter of best practice.

**C6: Uncertainty and contestability.** Epidemiologic priors come with confidence intervals, biomarker thresholds evolve with the literature, and oncologists may legitimately recommend treatment outside guideline parameters for refractory patients with no standard of care remaining, patients with contraindications to standard therapy, or other nuanced clinical situations requiring judgment. A runtime, therefore, cannot enforce a hard refusal uniformly: it must distinguish hard contraindications from soft norms while treating overrides as first-class auditable events. We draw the line explicitly. A *justified override* is a clinician-authored deviation from a *soft* norm (C6), attached to a recorded rationale and a responsible signer, that the specification permits and the audit layer preserves; it is a legitimate exercise of clinical judgment, not an error. An *agent failure*, by contrast, is a violation of a *hard* constraint (a C1 contraindication, a C2 coding error, a C4 interaction) that the runtime should have blocked, or an override lacking an authorized signer and rationale. The runtime's job is thus not to prevent all deviation but to guarantee that every deviation is classified, attributed, and logged, so that a justified override and a suppressed failure can never look alike after the fact. Recent clinical work on reasoning-oriented LLMs in medicine, RCTs of LLM influence on diagnosis, and analyses of consumer-LLM sycophancy under direct patient queries [21, 27, 82] all reinforce this; probabilistic and uncertain databases [22, 33] formalize the soft, judgment-bearing side of this lattice, and calibrating LLM agent confidences against such a formalism remains an open problem.

Clinical practice already operates within this lattice; what is missing is a runtime that enforces it on every agent action. Each constraint class above maps onto data-management primitives the community has already built and validated outside the clinical setting, and composing them, rather than scaling models, is what safety now depends on.

## 3 PRIMITIVES AS THE TRUSTED BASE

**Semantic integrity constraints (SICs).** Lee et al. [40] proposed a declarative framework for enforcing value, type, dependency, cardinality, and temporal constraints in AI-augmented pipelines. SICs cover most of C1, C2, and C4, while constraint-driven cleaning systems [7, 59, 64] extend the same principles to noisy upstream sources. For clinical use, what remains missing is integration with description-logic axioms over SNOMED class hierarchies, compilation strategies that determine whether a check belongs to pre-condition pruning vs. post-condition validation, and cost-aware ordering when checks have asymmetric latency and refusal costs. The longitudinal constraints of C4 push the framework further still: a rule such as "no renally-cleared drug at standard dose once eGFR falls below threshold" must compile into a dependency between the agent's proposed order and the most recent laboratory tuple, rather than a static value check.

**Semantic operators with accuracy guarantees.** LOTUS [53] and Palimpzest [45] show that LLM-call-based operators can be optimized with quality and cost guarantees, and CAESURA [77] extends this to multi-modal sources; data agents have since become a first-class community topic, reframing databases as a substrate for LLMs [25, 31, 43, 47, 51, 75]. Clinical workloads demand heterogeneous cost models, criticality-dependent guarantees, and constraint-driven operator reordering, for which constraint-driven NL2SQL [44, 60, 87] and clinical NL2SQL benchmarks [41] offer aligned starting points to build on. The distinguishing demand is that operator selection be criticality-aware, so that the planner never trades accuracy for latency on a contraindication check the way it might on a background literature summary.

**Provenance and lineage.** The semiring framework by Green, Karvounarakis, and Tannen [14, 29] provides a compositional algebra for tracking why each tuple in a query result was produced. Systems such as URSPRUNG [62], DPDS [13], and Smoke [56] extend the idea to data-science pipelines through automatic fine-grained lineage capture, while mlinspect and ArgusEyes [28, 63] screen ML pipelines using lineage as the substrate. These jointly provide a natural foundation for C3. The clinical setting, however, imposes stricter requirements: provenance must persist across federation boundaries, remain interpretable to non-technical reviewers in human-readable form, and be authenticated, since audit-log tampering is among the simplest attacks against deployed clinical agents [19]. A subtler requirement is that provenance survive summarization: when an LLM operator condenses several records into a single claim, the lineage of that claim must still resolve to the individual source tuples an auditor can re-examine.

**Federated query under access control.** SMCQL, Conclave, and Senate [4, 55, 81] demonstrate secure SQL execution over federated relational data, with motivating applications drawn explicitly from healthcare. Hippocratic Databases [1] and Qapla [49] formalized policy compliance for database-backed systems before the agent era, while the OHDSI ecosystem [58] and FHIR-Ontop-OMOP [83] demonstrate clinical-vocabulary and legal harmonization across institutions at scale. The open question is composition: enforcing the join of the policy lattice must rest with the planner, not the individual tools whose paths an agent might traverse.

**Probabilistic and uncertain databases.** ActivePDB and its predecessors, including MCDB [22, 33], provide a formal foundation for C6, including probabilistic constraints, Bayesian priors over biomarker thresholds, and confidence-bearing claims, whose connection to LLM agent outputs remains largely unexplored. Closing this gap means treating an operator's confidence as a probabilistic annotation that composes with the hard constraints, so that a soft threshold from C6 is evaluated explicitly and, once breached, surfaced as a calibrated warning rather than passing silently.

**Materialized views over evolving guidelines.** Clinical guidelines evolve irregularly: NCCN topics undergo multiple revisions per year, and ESMO has introduced "Living guidelines" for selected tumor types. A constraint-first runtime that compiles guideline excerpts into views must therefore support incremental view maintenance; modern IVM substrates such as DBSP [9] and DBToaster [37] apply directly to guideline-as-views compilation.

**Authoring and validating the specification.** A fair objection is that guidelines are not code: they contain ambiguity, exceptions, evolving recommendations, and passages that call for expert judgment, so encoding them as rigid rules risks a spurious sense of precision. Our design confronts this rather than assuming it away. First, only the *computable subset* is compiled: the near-deterministic contraindications and bundle steps of C1, with genuinely judgment-bearing passages left to the soft-norm and uncertainty machinery of C6 rather than forced into hard rules. Second, we do not expect each institution to formalize guidelines from scratch; the computer-interpretable-guideline community has spent two decades building formalisms and curation processes (GLIF, PROforma, FHIR Clinical Reasoning + CQL [6, 54, 70, 71]), which we reuse as *community-maintained, version-pinned "guideline-as-code" packages*. Third, validation becomes a first-class data-management task: a candidate package is a query one can test against retrospective cohorts, differentially compare across versions, and hold accountable through the same provenance and audit trail (C3) that governs live decisions, so that a mis-encoding surfaces as a measurable false-refusal or false-permit rate rather than a silent error. Curation, coverage, and drift of these packages at scale remain open (§5).

## 4 A CONSTRAINT-FIRST RUNTIME

The runtime has four layers (specification, planner, execution, audit), each exposing clinical concerns to clinicians and data concerns to data engineers behind deliberately narrow inter-layer interfaces.

**Specification** combines several layers: a SIC-style relational fragment [40], which LLM-assisted schema auditing [65] can help bootstrap; SHACL/OWL fragments over relevant ontologies; a Datalog-like rule layer for guideline excerpts curated as community-maintained guideline-as-code packages (version-pinned per institution, following HL7 FHIR Clinical Reasoning + CQL [6, 54, 70, 71]);

and a policy DSL naming the access scope of each agent capability [1, 49], a scope the planner may narrow but never widen.

**Planner** compiles the prompt with the active specification into a constrained DAG of tool calls (EHR queries, PrimeKG [12] and DDInter [84] lookups, literature retrieval, semantic-operator LLM calls), interleaved pre/mid/post-condition checks, and enforcement actions (refuse, warn-and-log, request-clinician-confirm). The cost model integrates classical query costs, LLM latency, and pricing [45, 53], per-tool capability scope [20], and a clinical-criticality dimension rendering contraindication checks non-skippable.

**Execution** wraps every tool call and LLM operator with automatic provenance capture, extending URSPRUNG-style techniques to agentic settings [13, 56, 62]. It coordinates federated access through OHDSI/FHIR endpoints [58, 83] with access-controlled query processing [4, 55, 81], and schema-validates tool outputs before downstream operators consume them [46, 61, 94]. Constraint violations interrupt execution and either suppress the failing branch or trigger a clinician override, recorded as a first-class event, extending human-in-the-loop task decomposition [90] to the agentic setting.

**Audit** emits, per agent decision, a replayable evidence dossier in the spirit of interpretable analytics for high-stakes clinical applications [89]: the active constraint set, the records and queries supporting each claim, LLM-operator outputs, constraint-trigger points, and clinician overrides, keyed by semiring-annotated provenance [29] and stored append-only with signed checkpoints in the spirit of in-toto and history-tree tamper-evident logging [19, 74]. Aligned with DECIDE-AI [80] and regulatory guidance [18, 73, 78], a tumor-board chair signs the dossier that an auditor replays.

*Overhead and trade-offs.* These layers are not free, and the design is deliberately shaped by their cost. Most constraint checks are cheap relational or ontology lookups that the planner pushes down and evaluates as preconditions, so they prune rather than add work; the expensive operators are the LLM calls the agent already makes. The cost model treats safety as a first-class dimension: contraindication checks (C1, C4) are marked non-skippable and paid regardless of latency, whereas soft checks are ordered by the same cost/quality optimization used for semantic operators [45, 53]. Provenance capture and append-only logging add storage and a bounded per-step write, the accepted price of auditability. The real trade-off is therefore not raw speed but *expressiveness versus enforceability*: constraints must be rich enough to catch real harms yet restricted enough to compile and check within the clinical decision cycle, and quantifying this frontier on logged MTB and ICU workloads is an explicit open question (§5).

*Position.* Most current clinical-agent prototypes [8, 17, 34, 85, 88] layer general-purpose agent frameworks atop an EHR query layer; the runtime we sketch sits beneath them, compiling tool calls and LLM operators into the constrained DAG, much as semantic-operator systems [45, 53] relate to ad hoc LLM pipelines.

## 5 OPEN RESEARCH QUESTIONS

**Q1. Heterogeneous constraint compilation.** How can a mixed specification (SIC [40] + SHACL/OWL + Datalog + probabilistic [22, 33]) compile into a single executable plan with provable enforcement? When do OWL axioms admit precondition pushdown rather than a dedicated reasoner call, and when do Datalog guideline rules such as the SEP-1 three-hour bundle [23] admit forbidden-tuple compilation versus runtime instantiation against the longitudinal patient record?

**Q2. Tamper-evident provenance and override semantics.** Define a provenance algebra in which clinician overrides are first-class events, queryable by auditors, statistically analyzable, and tamper-evident under an honest-but-curious storage adversary, building on the semiring framework [29], in-toto [74], history trees [19], and provenance-driven security analysis [50]. The conceptual challenge is defining a *justified* override: one that preserves clinical judgment while remaining automatable against the specification and aligned with emerging regulatory expectations [18, 26, 73, 78] for non-deterministic clinical software.

**Q3. Cost-aware planning with clinical criticality and capability scope.** Extend semantic-operator optimization [45, 53] with a criticality dimension, asymmetric refusal costs, and a capability-scope term penalizing tools whose access privileges exceed the active prompt's needs. Ground the cost model in logged MTB and ICU workloads, where clinical outcomes and time-to-treatment are already instrumented as operational signals [38, 66]. A subtlety is that criticality is not a fixed per-constraint weight: the same check may be non-skippable inside an ICU sepsis bundle, yet advisory at an outpatient follow-up, so criticality must itself be a function of the clinical context that the planner reads from the specification.

**Q4. Capability-scoped federated execution.** Compose institutional access policies across an agent's execution trajectory spanning OMOP/FHIR federations and secure federated query backends [4, 55, 58, 81]; plans whose composed access scope exceeds the permissions granted by any participating institution should be rejected at planning time rather than at execution time. This calls for a compositional semantics of access scope, in which the scope of a plan is derived from the scopes of its operators and checked against each institution's grant before any tuple crosses a federation boundary, rather than being audited only after the fact.

**Q5. Incremental compliance views under guideline updates.** Treat versioned guideline packages as base relations and active prompts as queries, maintaining compliance under version changes through incremental view maintenance [9, 37]. Pinning the compliance view active at the time of a past decision is itself a security property that auditors must reconstruct. The open challenge is to bound the recomputation a revision triggers, so that compliance can be re-established online for the affected plans without recompiling every active plan from scratch.

## 6 CONCLUSION

A clinical agent's safety, security, and contextual intelligence come from the plan it compiles, not from the model that proposes it. The building blocks already exist in data management and clinical informatics: a runtime can compile each agent action into a constrained, provenance-tracked, and capability-scoped query plan, then emit a dossier that the attending clinician signs and an auditor can replay. This extends the database community's long-standing role as a trusted computing base to language-model agents, and lets

us inspect an agent's behavior in its dossier instead of inferring it from a model card. The six constraint classes we catalog and the primitives we map them onto are not a wish list but an inventory of components the community has already built, tested, and deployed outside medicine; what is new is the demand that they compose into a single compiled artifact whose guarantees a clinician can sign and an auditor can replay. The bedside merely forces the question first; the same substrate fits any setting where agents act over governed, uncertain data under consequential rules. Realizing it will require the clinical and data-management communities to co-design the specification language, the cost model, and the audit format together, which is exactly the kind of collaboration this workshop exists to foster. A natural first step is not a larger clinical model but a minimal end-to-end slice of this runtime (one constraint class, one federation boundary, one signed dossier) exercised on a real molecular-tumor-board or ICU workload and measured for what it costs and what it catches. The next advance at the bedside is thus not a better model but a runtime that compiles whatever model we use into plans we can audit.

## 7 AUTHORS

**Kaiping Zheng (data management).** Senior Research Fellow at the School of Computing, National University of Singapore (NUS), working at the intersection of data management, machine learning, and clinical medicine. Her research develops robust medical data-management methods that turn noisy electronic health records into trustworthy clinical signals, with responsible-AI techniques aligned with clinical reasoning.

**Horng-Ruey Chua (biomedical).** Head and Senior Consultant in the Division of Nephrology, National University Hospital (NUH), and Adjunct Associate Professor at NUS Yong Loo Lin School of Medicine (YLLSoM).

**Si Wei Kheok (biomedical).** Senior Consultant in Diagnostic Radiology at Singapore General Hospital, Director of Head and Neck Radiology and of MRI Patient Care Services, and Deputy Head of Neuroradiology, with appointments at Duke-NUS, NTU, and NUS YLLSoM.

**Kee Yuan Ngiam (biomedical).** Head and Senior Consultant in the Division of General Surgery (Endocrine and Thyroid Surgery), Department of Surgery, NUH; Senior Consultant in Surgical Oncology at the National University Cancer Institute, Singapore (NCIS); and Adjunct Professor of Surgery at NUS YLLSoM.

**Marcus Chun Jin Tan (biomedical).** Consultant in Ophthalmology at NUH and Deputy Group Chief Technology Officer of the National University Health System.

**Robert John Walsh (biomedical).** Consultant in Haematology-Oncology at NCIS.

## ACKNOWLEDGMENT OF GENERATIVE-AI USE

The illustration in Figure 1 was produced with a text-to-image model from an author-written prompt and then curated by the authors. Large language models were also used to assist with copy-editing and to smooth prose. All technical claims, the framework, the six constraint classes, and the overall research agenda are entirely the authors' own original work.

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
