# OpenReview forum: "From Guardrails to Compilers: A Constraint-First Runtime Substrate for Clinical AI Agents"
_VLDB.org/2026/Workshop/BioDMS — BioDMS 2026 ProjectTalk_

### Official Review · Reviewer_GJdT · 2026-06-04

**Summary:**

The paper presents a comprehensive plan for developing a constraint-first runtime that compiles every agent action into a  provenance-tracked query plan. The authors argue that the problems with LLMs are runtime failures, and not model failures, and that data primitives for constraint enforcement and provenance tracking will improve the results of agents and make them suitable for clinical deployment.

**Confidence Of Review:**

3

**Detailed Feedback Points:**

S1: Great breakdown of the constraints which come from the clinical side, and the primitives which come from the data management side.

S2: This is a very ambitious project with lots of potential for collaborations.

W1: The abstract needs another pass, because it usually doesn't contain references. It should be mostly about what you are doing, how and the results, as a brief summary of the entire paper.

W2: The paper is quite hard to read, due to the domain-specific language, and convoluted sentences.

**Relevance For Biodms:**

4

---

### Official Review · Reviewer_UEhw · 2026-06-08

**Summary:**

Dear Authors,

I have reviewed your manuscript entitled “From Guardrails to Compilers: A Constraint-First Runtime Substrate for Clinical AI Agents.” The authors describe a potential constraint-first runtime architecture for clinical AI agents, claiming that many safety, security, and reliability issues of current LLM-based systems should be addressed at the runtime level rather than through guardrails. The authors propose a constraint-first runtime that compiles every agent actions into constrained and provenance-tracked query plans that integrate semantic integrity constraints, clinical guidelines, ontological axioms, provenance mechanisms, federation-aware access policies, and auditing capabilities. The manuscript further outlines a four-layer runtime architecture (specification, planning, execution, and audit), and discusses several open research questions.

**Confidence Of Review:**

3

**Detailed Feedback Points:**

1) The paper presents a compelling and timely vision for reframing clinical AI safety as a data-management problem. The manuscript is clearly written, highly relevant to the BioDMS audience, and supported by an extensive review of both the clinical AI and data-management literature.

2) The proposed framework integrates a broad range of concepts, including semantic constraints, provenance, federated query processing, access control, and guideline-aware reasoning, into a unified runtime architecture. This perspective is interesting and has the potential to stimulate discussion within the community.

3) A major limitation is that the manuscript remains largely conceptual. Most of the proposed architecture is just presented, and many of the key technical challenges are deferred to the open research questions section. As a result, it is difficult to assess the practical feasibility of the proposed runtime and the extent to which it can be realized in real-world clinical settings.

4) The framework relies heavily on the assumption that clinical guidelines can be translated into executable “guideline-as-code” specifications. However, many clinical guidelines contain ambiguity, exceptions, evolving recommendations, and expert judgment that are difficult to encode as rigid logical rules. The paper would benefit from a deeper discussion of how these specifications would be created, maintained, and validated at scale.

5) The discussion of “soft norms” and clinician overrides highlights an important challenge. The manuscript argues that clinicians must be able to override recommendations in certain situations, yet the conceptual distinction between a justified clinical override and a failure of the underlying agent remains unclear. Since overrides play a central role in the proposed framework, further clarification would strengthen the paper.

6) The proposed architecture introduces numerous additional runtime components, including constraint checking, ontology reasoning, provenance tracking, and policy enforcement. While these mechanisms may improve safety and auditability, the manuscript does not discuss the practical costs associated with these operations or the trade-offs that may arise between safety guarantees, system complexity, and deployment in real-world clinical environments.

**Relevance For Biodms:**

3

---

### Official Review · Reviewer_aiBL · 2026-06-11

**Summary:**

This paper explores, what they refer to as a conceptual shift from guardrail-based approaches to more structured, compiler-inspired methods for managing complex workflows, with a focus on biomedical and clinical data settings.

**Confidence Of Review:**

2

**Detailed Feedback Points:**

1. Relevance and Topic Importance: The topic of the paper is clearly relevant to the BioDMS workshop, particularly in its focus on improving reliability and structure in workflows that intersect with biomedical data processing. The overall direction is interesting and aligns well with the goals of the workshop.

2. Clarity and Accessibility: From a data management perspective, the paper is quite difficult to follow. The concepts and terminology are not always introduced in a way that is accessible to readers outside the immediate subcommunity, and the lack of clear, concrete examples makes it challenging to build intuition about the proposed ideas. Improving the clarity of presentation, e.g., by adding a running example and/or more explicitly connecting abstractions to familiar data management concepts, would significantly enhance readability and accessibility. More generally, even after a second read, it remains unclear what the core objective or primary goal of the paper is.

3. Connection to Data Management and Biomedical Use Cases: While the paper is relevant by topic, the connection to both data management and biomedical use cases could be made more explicit and concrete. It is not always clear how the proposed ideas translate into specific challenges or opportunities within biomedical data systems, or how they relate to established data management problems (e.g., query optimization, workflow management, or data integration). In addition, the paper would benefit from a more concrete articulation of its contributions (e.g., as traditionally introduced as the end of the introduction)

4. Author Introductions: The paper does not state the authors’ communities as requested in the CFP, and it is not clear which authors contribute the data management perspective in this project.

5. Use of AI-Generated Content: Several parts of the paper appear to be AI-generated without a clear acknowledgement. For example, Figure 1 appears to be AI-generated and is somewhat overwhelming from a reader’s perspective. Portions of the text also read as highly stylized or generated (e.g., "Clinical reasoning at the bedside, on the wards, and at the tumor board operates against a structured constraint lattice that today’s agents rarely respect."). While the use of AI tools may be acceptable (as far as I am aware), it should be clearly acknowledged.

**Relevance For Biodms:**

2